# Facile Synthesis of Aminated Graphene Quantum Dots for Promising and Selective Detection of Cobalt and Copper Ions in Aqueous Media

**DOI:** 10.3390/molecules27227844

**Published:** 2022-11-14

**Authors:** Weitao Li, Ningjia Jiang, Luoman Zhang, Yongqian Chen, Jie Gao, Jihang Zhang, Baoshuo Yang, Jianxin He

**Affiliations:** 1Textile and Garment Industry of Research Institute, Zhongyuan University of Technology, Zhengzhou 450007, China; 2Institute of Nanochemistry and Nanobiology, School of Environmental and Chemical Engineering, Shanghai University, Shanghai 200444, China

**Keywords:** graphene quantum dots, Cu^2+^, Co^2+^, amino functionalization

## Abstract

Due to the rapid development of industrialization, various environmental problems such as water resource pollution are gradually emerging, among which heavy metal pollution is harmful to both human beings and the environment. As a result, there are many metal ion detection methods, among which fluorescence detection stands out because of its rapid, sensitive, low cost and non-toxic characteristics. In recent years, graphene quantum dots have been widely used and studied due to their excellent properties such as high stability, low toxicity and water solubility, and have a broad prospect in the field of metal ion detection. A novel high fluorescence Cu^2+^, Co^2+^ sensing probe produced by graphene quantum hydrothermal treatment is reported. After heat treatment with hydrazine hydrate, the small-molecule precursor nitronaphthalene synthesized by self-nitrification was transformed from blue fluorescent GQDs to green fluorescent amino-functionalized N–GQDs. Compared with other metal ions, N–GQDs are more sensitive to Cu^2+^ and Co^2+^ on the surface, and N–GQDs have much higher selectivity to Cu^2+^ and Co^2+^ than GQDs. The strategy proposed here is simple and economical in design.

## 1. Introduction

With the deterioration of land, environment and water quality caused by rapid industrialization, the global demand for clean environment, geology and water is increasing day by day, and it is urgent to detect, remove or recover unfavorable metal ions to meet the demand for environmental quality [1]. These industrial effluents or pollutants containing heavy metal ions, inorganic substances and organic matter enter freshwater bodies and pollute freshwater systems, which has become a major global environmental threat [2]. Among them, heavy metal wastewater (containing Cd^2+^, Co^2+^, Cu^2+^ and other heavy metal ions) is among the most serious industrial wastewater to water and environment and the greatest harm to human beings, and the common water treatment methods [3] cannot decompose and destroy the heavy metals in the wastewater.

Excessive levels of Cu^2+^ and Co^2+^ can cause heavy metal poisoning once they enter the bloodstream [4]. Cu mainly causes protein denaturation in the body, and another toxic manifestation is damage to red blood cells, resulting in hemolysis and anemia [5]. The clinical manifestations of mild cobalt poisoning are loss of appetite, vomiting, diarrhea, etc., and serious ones can cause sensorineural deafness and optic nerve atrophy [6]. At this time, it is necessary to develop an accurate, sensitive, simple and rapid metal ion detection technology.

In recent decades, various methods for detecting heavy metal ions have been developed [7]. These include atomic absorption spectrometry [8], electrochemical methods [9], X-ray fluorescence spectrometry [10], and the fluorescence method [11]. Among them, the fluorescence method is based on the fluorescence intensity of metal ions, enabling selective detection of metal ions, with high selectivity, high sensitivity, low cost, simple operation and other unique advantages.

In recent years, many fluorescent nano materials [12,13,14,15], because of their unique optical properties of applications in biological [16,17,18,19,20,21,22] or chemical [23,24,25] sensors, and graphene quantum dots (GQDs) [26,27], as a zero-dimensional material, with good biocompatibility and low toxicity, solubility in water, high stability and other unique features, have great potential in the field of heavy metal ion detection. GQDs are generally synthesized by top-down [28,29] or bottom-up [30,31,32,33] methods. Compared with the top-down approach, the use of smaller molecules as starting materials provides a more controlled strategy for experiments, more control over optical properties, and higher yields and good carbonation [34,35,36]. Additionally, most GQDs will inevitably introduce surface defects, such as hydroxyl and carboxyl groups and other oxygen-containing groups, in the preparation process, and these oxygen-containing defects form non-radiative complexes of electron-hole pairs, resulting in low quantum yields of GQDs. The amino-functionalized GQDs can passivate the surface of GQDs, introduce functional groups or react with oxygen-containing groups on the surface, so as to inhibit the carrier non-recombination process. Amino functionalization modification can also change the emission peak position of GQDs.

A novel fluorescence probe based on GQDs hydrothermal treatment is reported, as shown in Figure 1. After heat treatment with hydrazine hydrate, blue GQDs were transformed into green N–GQDs. Compared with other transition metal ions, Cu^2+^ and Co^2+^ on the surface of N–GQDs have higher binding affinity and faster chelation kinetics, and the selectivity of N–GQDs to Cu^2+^ and Co^2+^ is much higher than that of GQDs. Therefore, using the above characteristics, we prepared a simple fluorescence sensor, which can detect Cu^2+^ and Co^2+^ in an aqueous solution.

## 2. Results and Discussion

Figure 1, showing the morphology of GQDs and N–GQDs, was analyzed by atomic force microscopy and transmission electron microscopy. As shown in Figure 1a,b, the average thickness of GQDs and N–GQDs is between 0–14 and 0–6 nm, respectively, and most of them are composed of multilayer graphene. Figure 1c,d shows that both GQDs and N–GQDs are nanodots with relatively uniform particle sizes and good dispersion, with average particle sizes of 1.5 and 2.8 nm, respectively. Figure 1e,f clearly shows the lattice fringes where GQDs and N–GQDs are located, indicating that both of them have high crystallinity. The lattice spacing of GQDs and N–GQDs is 0.22 and 0.23 nm, respectively, corresponding to the (100) and (110) planes of graphene, respectively [37].

Figure 2 shows the XPS spectrum of N–GQDs. Figure 2a shows that there are 80.59% C, 13.19% O and 6.23% N in N–GQDs. The deconvolution of C1s peaks in Figure 2b shows four characteristic peaks of C–C, C–N, C–O and C=O at 284.58, 284.68, 285.68 eV and 286.38 eV. As shown in Figure 2c, three types of N phase keys can be observed, including NH_2_ (398.98 eV), N–C (399.28 eV), and N–O (400.93 eV). O1s spectrum (Figure 2d) at 530.88, 532.38 and 533.18 eV, and the binding energies correspond to O–C, O=C and O–N, respectively. XPS data verified that the GQDs surface contains the NH_2_, –OH, –COOH, and –N–OH functional groups [38].

Figure 3a,b shows UV–visible absorption spectra of GQDs and N–GQDs in an aqueous solution, fluorescence spectra at 311 and 410 nm excitation, respectively, and emission spectra at 482 nm wavelength, respectively. From the ultraviolet absorption spectrum in the figure, the absorption region of N–GQDs extends from the ultraviolet region to the visible region, with the longest wavelength up to 650 nm. GQDs extends up to 490 nm. The illustrations in Figure 3a,b are GQDs and N–GQDs under natural light and UV light, respectively. Figure 3c,d shows the fluorescence spectra of GQDs and N–GQDs at different excitation wavelengths, respectively, as shown in the figure, the fluorescence intensity of GQDs and N–GQDs increases with the increase in excitation wavelength at the beginning, and then reaches the maximum fluorescence at the excitation wavelength of 311 and 410 nm, respectively, and the excitation wavelength continues to increase, the fluorescence intensity of GQDs and N–GQDs decreased. The best excitation wavelength of GQDs and N–GQDs is 311 and 410 nm, respectively, which is corresponding to Figure 3a,b.

Figure 4 shows the fluorescence intensity of N–GQDs at different pH values. As shown in Figure 4, the initial pH value of N–GQDs is 8. It can be observed that the fluorescence intensity decreases with the decrease in pH value, and plummets when pH decreases to 5. However, with the increase in pH, the fluorescence intensity gradually decreases, and it can be concluded that N–GQDs are resistant to strong alkali, but not to strong acid. In addition, the time stability experiment and dispersion stability experiment of GQDs and N–GQDs are performed, respectively. According to Appendix A, both kinds of quantum dots have good stability, but the dispersion stability of N–GQDs is better than that of GQDs.

Figure 5a shows the fluorescence intensity histograms of GQDs and N–GQDs in different metal ionic solutions. Figure 5b corresponds to N–GQDs with different metal ions in Figure 5a, which is the fluorescence spectra of N–GQDs with different metal ions. As shown in Figure 5a, N–GQDs have higher detection sensitivity than GQDs for Cu^2+^ and Co^2+^, and obvious fluorescence quenching phenomenon occurs. Additionally, according to Appendix A, the concentration of Ni^+^ and Zn^2+^ does not have a good linear relationship with the intensity difference ratio ((F_0_ − F)/F_0_) of N–GQDs. So it can be used for the detection of Cu^2+^ and Co^2+^. Due to the reaction between hydrazine hydrate and carbonyl group on GQDs, GQDs can be effectively reduced when the amino group is functionalized, thus promoting the red shift of GQDs. Amino-functionalized graphene quantum dots (N–GQDs) have better luminescence performance and better fluorescence stability than conventional graphene quantum dots (GQDs). Figure 5c,d shows the pictures of GQDs and N–GQDs under natural light and ultraviolet light. The light color of N–GQDs in different metal ion solutions is different.

It can be observed from Figure 6 that when the molar concentrations of Cu^2+^ and Co^2+^ are in the range of 0.2–1 µM, there is a good linear relationship between the molar concentrations of Cu^2+^ and Co^2+^ and the quenching efficiency (F_0_ − F)/F_0_, where F_0_ and F are the fluorescence intensities at 410 nm excitation in the absence and presence of Cu^2+^/Co^2+^, respectively. With the increase in Cu^2+^/Co^2+^ concentration, the fluorescence intensity of N–GQDs gradually decreases. The regression equations of Cu^2+^ and Co^2+^ are y = −0.0106x + 0.1042 and y = 0.0169x − 0.0017, respectively. In addition, the fluorescence recovery experiment of N–GQDs containing Cu^2+^/Co^2+^ was performed by adding EDTA. According to Appendix A, the fluorescence intensity of N–GQDs containing Cu^2+^/Co^2+^ can be restored by adding EDTA.

In addition, the content of Cu^2+^ was determined by iodine quantity method. In weakly acidic solution, Cu^2+^ could be reduced to CuI by KI and a certain amount of I_2_ was precipitated (2 Cu^2+^ + 4 I^−^ →2 CuI + I_2_). The precipitated I_2_ was titrated with Na_2_S_2_O_3_ standard solution and starch was used as the indicator. The content of copper was indirectly measured (I_2_ + 2 S_2_O_3_^2−^ → 2 I^−^ + S_4_O_6_^2−^). It is found that this method is more suitable for the determination of high copper content, and N–GQDs can detect the content of trace Cu^2+^. According to the Groundwater Quality Standard (submitted for approval), the content of copper in class Ⅱ water of the groundwater quality limit should be ≤0.05 mg/L, which is within the Cu^2+^ detection range of N–GQDs. Additionally, because of the complicated experimental process, the error is large. Its solution environment also needs to be weak acid conditions compared with N–GQDs, it is more restrictive.

As shown in Figure 7a,b, the crystal plane spacing of [002] of GQDs and N–GQDs is basically the same, 3.64 Å and 3.43 Å, respectively. The results are similar to those reported in the literature. Figure 7c,d shows the infrared spectra of GQDs and N–GQDs, from which the existence of the amino and hydroxyl functional groups can be observed. In Figure 7c, there are two broadband absorption bands at 3209 and 3063 cm^−1^, which are caused by the stretching vibration of O–H and N–H, respectively. The peaks at 1538, 858 and 762 cm^−1^ were C–O, C–OH and C–N vibrations, respectively. In Figure 7d, there are two broadband absorption bands at 3345 and 3195 cm^−1^, which are caused by the stretching vibration of O–H and N–H, respectively. The peaks at 1626, 984 and 804 cm^−1^ were C–O, C–OH and C–N vibrations, respectively. In addition, the stretching vibration of C=C at 1359 and 1389 cm^−1^ is observed for GQDs and N–GQDs, respectively [38].

Figure 8 shows the Raman spectra of GQDs and N–GQDs. As shown in the figure, there are two obvious peaks of GQDs and N–GQDs at approximately 1300 and 1600 cm^−1^, respectively, corresponding to the D-peak and the G-peak in the Raman spectrum of graphene quantum dots, respectively. The intensity ratio of the D-peak to the G-peak of N–GQDs ranges from 0.98 to 1.04, indicating that N–GQDs are well crystallized.

## 3. Materials and Methods

### 3.1. Chemicals and Reagents

Chemicals were purchased and used directly without further purification. Concentrated sulfuric acid, concentrated nitric acid, naphthalene, hydrazine hydrate, etc., were purchased from Sinopharm Group.

### 3.2. Apparatus and Characterization

The height map of GQDs was measured by atomic force microscopy (FMP-3D Infinity, Oxford, UK), the characteristic peak of GQDs was measured by confocal Raman spectrometer (Horiba Xplora Plus, Shanghai, China), and the infrared spectrum of GQDs was measured by a Fourier transform infrared spectrometer (Bruker TENSOR37, Shanghai, China). Transmission electron microscope (JEM-2010F, Japan Electronics Co., Ltd., Tokyo, Japan) obtained high-resolution TEM images, an X-ray diffractometer (Philips X ‘Pert Pro, Shanghai, China) obtained XPS atomic measurement spectrograms, fluorescence spectra were measured with a fluorescence spectrometer (RF-6000, Shimadzu, Shanghai, China) and UV spectra were measured with a UV—visible near-infrared spectrometer (Cary5000, Agilent, Beijing, China).

### 3.3. Synthesis of Nitronaphthalene

An amount of 0.2 g of naphthalene was added to a mixed solution of 160 mL of concentrated nitric acid and concentrated sulfuric acid (3:1), resulting in a nitration reflux reaction at 80 °C for 48 h, and was then taken out after cooling. The obtained product was slowly added to 2000 mL deionized water, then pumped and filtered with a filter membrane to remove the acid solution, washed until the filter solution was neutral, then dried at 80 °C for later use, and then the nitronaphthalene was obtained [19,36].

### 3.4. Synthesis of GQDs and N–GQDs

GQDs and N–GQDs were synthesized by a solvothermal method using nitronaphthalene as a precursor.

N–GQDs were prepared by a hydrothermal reaction [17,21,22,25,33,34] reported in the literature. An amount of 0.4 g nitronaphthalene was dissolved in a mixture of 4 mL hydrazine hydrate and 36 mL deionized water, and was dispersed by ultrasound for 10 min to make it uniform. The solution was then transferred to a 50 mL Teflon lined reactor, heated to 180 °C for 12 h, and cooled to room temperature naturally. The insoluble substance in the product was removed by filtration with a 0.22 μm microporous membrane. The resulting quantum dot solution was then placed into a dialysis bag for dialysis, and the resulting N–GQDs solution was placed into a 60 °C oven for drying to obtain solid powder for further experimental characterization.

GQDs were prepared by adding 0.4 g of nitronaphthalene to 40 mL of deionized water. The subsequent experimental steps are consistent with N–GQDs.

### 3.5. Ion Detection Experiment

Different metal ion solutions (Mn^2+^, Cd^2+^, Ni^+^, K^+^, Al^3+^, Cu^2+^, Na^+^, and Co^2+^) of 10 mM were prepared, and 500 μL of GQDs/N–GQDs was mixed with 500 μL of deionized water as a blank control. In the experimental group, 500 μL of GQDs/N–GQDs was mixed with 500 μL of different metal ion solutions, and the fluorescence intensity values were measured and compared. EDTA with the same molar mass and volume as metal ion solution was added to the mixture of N–GQDs and metal ion solution, and the cycle was repeated three times to measure the fluorescence intensity, respectively.

Cu^2+^ and Co^2+^ solutions with different molar concentrations were prepared, i.e., 0, 0.2, 0.4, 0.6, 0.8 and 1 μM, mixed with GQDs and N-GQDs, respectively, and their fluorescence intensity was measured.

## 4. Conclusions

In conclusion, we have developed a new type of luminescent quantum dots through the autonomous synthesis of the precursor nitronaphthalene. By amine functionalization, the graphitization degree of N–GQDs was obviously improved, and it had good fluorescence stability. In addition, N–GQDs show sensitive selectivity for metal ions, and obvious fluorescence quenching occurs in Cu^2+^ and Co^2+^ solutions. In the future, we will explore the application of this method in ion detection sensors and other fields.

## Data Availability

The data can be made available upon reasonable request.

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
