# Peer review of "Facile Synthesis of Aminated Graphene Quantum Dots for Promising and Selective Detection of Cobalt and Copper Ions in Aqueous Media"

_molecules, 2022, doi:10.3390/molecules27227844_

Round 1

Reviewer 1 Report

The reviewed manuscript molecules-1917977 reports a method for detection of cobalt and copper ions in aqueous media by using aminated graphene quantum dots. There are many obvious mistakes in the text and Figures.

Comments:

Ø  The authors mentioned “atomic fluorescence”, in fact, the method used in this work was not atomic fluorescence spectrometry, but molecular fluorescence spectrometry. It is misleading in the text.

Ø  atomic fluorescence” mentioned in Abstract and Introduction, should be “atomic fluorescence spectrometry”, please revise.

Ø  Line 5, Para 3, in Section “1. Introduction”. Please explain “atomic fluorescence light”.

Ø  Line 10, Para 1, in Section “2.2. Apparatus and characterization”. The model of the fluorescence spectrometer should be added.

Ø  Line 1, Para 2, in Section “2.4. Synthesis of GQDs and N-GQDs”. Some references should be cited here.

Ø  Line 6, Para 1, in Section “3. Results and Discussion”. “Figure 1b-c” should be “Figure 1c-d”.

Ø  Figure 5. The figure caption is very confused. What did the author want to demonstrate through Figure 5b? From Figure 5b, it could be concluded that the proposed N-GQDs could not utilize to construct an analytical method for detection of Co and Cu at all.

Ø  Figure 5a. How to obtain that N-GQDs has a better detection sensitivity than GQDs from Figure 5a? The responses are very confused.

Ø  Figure 5a. N-GQDs also has obvious fluorescence quenching in the solution containing Ni and Zn, how to ensure the selectivity of the proposed method?

Ø  Figure 5c and 5d. How to tell images of GQDs and N-GQDs?

Ø  Figure 6. The value on the y-axis should be shown. The figure caption was wrong, “at different molar concentrations of N-GQDs”? The fluorescence intensity ratio (F0/F), please define F0 and F. Since fluorescence quenching occurred with the existence of Co and Cu, why were there negative slopes for the fitting curves? The linear range of 0-1 µmol is not correct, what is the lower limit of the linear range? 0.2 µmol? If yes, it was a very narrow linear range.

Ø  Experimental details should be added in all the figure captions.

Ø  Maybe, comprehensive interference study, comparison with similar methods, and sample analysis should be carried out in this work.

Ø  The English of this paper should be reconsidered.

Author Response

Thanks a lot for your valuable review. We have responded to the questions one by one, please see the attachment for details.

Reviewer 2 Report

In this paper, the authors reported a fluorescence sensing probe based on GQDs, which displayed high binding affinity and faster chelation and selectivity to Cu2+ and Co2+. After heat treatment, the obtained N-GQDs showed excellent selectivity to Cu2+ and Co2+, which is a very interesting experimental phenomenon. However, some minor problems should be solved in this paper before it is published in Molecules. Below are my comments:

1.    The authors mentioned that after heat treatment, the obtained N-GQDs displayed excellent selectivity to Cu2+ and Co2+. However, why do N-GQDs exhibit excellent activity after heat treatment? The mechanism should be discussed in the revised manuscript.

2.    Whether the surface structure of GODs has changed after heat treatment? The authors need to make a comparison, which may help explain the mechanism.

3.    The quality of Fig.2, Fig.3, and Fig.7 is too poor, please modify in the revised manuscript

4.    It is necessary to cite new and high-quality literature to enhance the persuasiveness of the paper:

Angew.Chem.Int.Ed. 2015, 54, 7176-7180

Angew.Chem.Int.Ed. 2018, 57, 9224-9237

Author Response

(The authors gave the same response as above.)
